# Detection of Hybrids in Willows (*Salix*, Salicaceae) Using Genome-Wide DArTseq Markers

**DOI:** 10.3390/plants13050639

**Published:** 2024-02-26

**Authors:** Radim J. Vašut, Markéta Pospíšková, Jan Lukavský, Jan Weger

**Affiliations:** 1Department of Biology, Faculty of Education, Palacky University Olomouc, 779 00 Olomouc, Czech Republic; 2Department of Botany, Faculty of Science, Palacky University Olomouc, 783 71 Olomouc, Czech Republic; 3Department of Phytoenergy, Silva Tarouca Research Institute for Landscape and Ornamental Gardening, Public Research Institute, 252 43 Průhonice, Czech Republic; 4The Nature Conservation Agency of the Czech Republic, Moravian-Silesian Regional Branch, 756 61 Rožnov pod Radhoštěm, Czech Republic

**Keywords:** *Salix*, willow, DArTseq, hybridisation, hybrid identification, systematics, biomass crop

## Abstract

The genus *Salix*, comprising some 400–500 species, is important in various alluvial or wet habitats of the northern hemisphere. It is a promising crop for applications such as biomass production, biofuels, or environmental projects. Clear species delimitation is crucial in ecology, biotechnology, and horticulture. DArTseq markers, a genome-wide technique, were tested for species and hybrid identification. A total of 179 willow samples were analysed, including six species of *Salix* subgen. *Salix* and four species of *Salix* subgen. *Vetrix*, including those used in biomass crop production, representing important European taxa. Identification of species-specific markers, clustering analyses (principal coordinate analysis, neighbor-joining) and Bayesian methods (Structure) unambiguously identified putative hybrids. In addition to demonstrating the high efficiency of DArT-seq markers in identifying willow hybrids, we also opened-up new questions about hybridisation processes and systematics. We detected unidirectional hybridisation between *S. alba* and *S. fragilis*, forming backcross hybrids, and we rejected the hypothesis that *S. fragilis* does not occur naturally in Europe. Further, the isolated position of *Salix triandra* within the genus was confirmed.

## 1. Introduction

Interspecific hybridisation plays an important role in plant evolution, promoting genetic diversity, adaptation to diverse environments, and the emergence of new species. Hybridisation occurs when two individuals of different species or varieties interbreed—differing in at least one hereditary trait—resulting in offspring that inherit a combination of traits from their parents [1] and contributing to their ability to thrive in varying conditions. It is a dynamic force influencing the genetic landscape and the evolutionary trajectories of plant species. Artificial hybridisation in agriculture and industry further accelerates these processes, leading to the creation of new crops with improved desirable traits. In agriculture, crop breeding is improving traits such as yield, disease resistance, and adaptability to different climates. In a changing environment, new hybrid varieties of woody species are becoming increasingly important as a source of renewable energy, for use in phytoremediation, for carbon sequestration, and other environmental benefits.

Some of the most promising crops for biomass production are hybrid poplars (*Populus* L.) and willows (*Salix* L.). Hybrid poplar and willow varieties are widely grown for biomass production, particularly for bioenergy and biofuel production, but also for furniture and bio-economy products with higher value; fast-growing hybrids have the ability to sequester carbon and to have an impact on water quality [2,3]. Willow shrubs have key characteristics for biofuels, bioproducts, and bioenergy—high sustainable yields in short rotation cycles, easy propagation from dormant hardwood cuttings, a broad underutilised genetic base, and the ability to resprout after harvest [4]. Due to their special physiological adaptations and ecological resilience, willows are promising plants for environmental projects such as ecosystem restoration, phytoextraction, rhizofiltration, and phytostabilisation [5].

In recent years, some negative aspects of the utilisation of allochthonous species in agriculture and landscape were also identified, including the risk of invasive behaviour. Although no willow species is currently included in the list of invasive species in Europe [6,7], some European species and their hybrids are considered a serious problem on other continents [8], negatively affecting riparian habitats [9,10,11]. Due to the difficulty of identifying willow hybrids, especially in young seedlings, reliable methods for identifying willow genotypes are therefore essential for assessing these risks of possible inbreeding of non-native species into valuable populations of native willows.

The genus *Salix* is the largest member of the family Salicaceae, with about 400–500 species listed [12,13]. Willows are exclusively trees and shrubs found mainly in the northern hemisphere, generally preferring humid or wet habitats—with tree species mainly confined to alluvial habitats, whereas shrub species often grow in wetlands or alpine-like habitats [14,15]. Clear species delimitation, together with knowledge of parental taxa in hybrids, is crucial for related fields of research such as ecology and plant science, biotechnology, or horticulture. Many controversies have arisen in the past due to the lack of clear species delimitation and often subjective nomenclature changes (without a solid background of various data). Classical systematics of willows, based solely on analyses of morphological variation, have often been severely biased by the facts that willows have apetalous flowers (and thus having a limited number of distinct reproductive characters), dioecy and phenology (flowering before leaf development), and large intraspecific phenotypic variability, especially in leaf shape variability. All these limitations are further complicated by hybridisation, which is generally considered to be quite common in the willow family.

The high number of described interspecific willow hybrids is probably overestimated, but an efficient tool for hybrid identification is awaited for the revision of hybridisation patterns in the genus. The recent application of genome-wide analyses using RAD (restriction site-associated DNA) sequencing has greatly helped in understanding hybridisation processes in alpine willows, determining gene flow and backcrossing in populations [16]. Alternative to RAD-seq is another a high-throughput genome-wide analysis method called Diversity Array Technology (DArTseq), which produces restriction site-associated SNP markers using a combination of a complexity reduction method and a next generation sequencing platform [17,18]. The power of DArTseq markers for genotypic identification of willow species was tested on four taxa (*S. alba* L., *S. triandra* L., *S. purpurea* L., and *S. viminalis* L.) widely distributed in central Europe, revealing that these evolutionarily divergent species are clearly genetically separated [19]. This suggests the potential usefulness of DArTseq markers for hybrid identification, which we are testing further in our study.

Two notorious critical hybrid complexes occur in Europe: *Salix alba*–*S. fragilis* L., and *Salix caprea* L.–*S. cinerea* L.–*S. viminalis*. Intensive hybridisation has been repeatedly documented for these species, using different types of molecular and cytogenetic markers such as RAPD, AFLP, cpDNA, microsatellite markers, or high number of SNPs [20,21,22,23,24,25]. In our study, we used DArTseq technology to test whether hybrids of these two most common hybrid complexes can be unambiguously recognised and the parental taxa conclusively identified. We aim to use this method in experimental crosses and breeding targeted to produce new native willow cultivars for agriculture and agroforestry systems and in field studies focusing on systematics or ecology including assessment of risks of inbreeding of allochthonous species for biomass production with autochthonous species.

## 2. Results

### 2.1. Frequency of the Species-Specific Markers

Co-analysis of the *CCV* (‘*caprea-cinerea-viminalis*’) and *AF* (*’alba-fragilis’*) data sets provided a total of 328,067 markers (SilicoDArT) and 205,110 markers (binary SNPs), respectively. The number of species-specific markers (markers; PA) varied between 7559 (*S. pentandra* L.; 2 samples) and 50,917 (*S. alba*; 27 samples) when counting at least one presence of the marker in the species but absent in all other taxa. To overcome the bias caused by the different numbers of samples per species, only markers present in more than half of the samples per species were considered. In total, this varied between 2365 (for *S. caprea*; 22 samples) and 24,525 (*S. alba*; 27 samples). The number of species-specific markers per sample varied on average between 1781 (*S. caprea*) and 21,234 (*S. triandra*). In general, species of *Salix* subgen. *Vetrix* (*S. caprea*, *S. cinerea* and *S. viminalis*) have a significantly lower number of species-specific markers than those of the subgenus *Salix* (summarised in Table 1).

#### 2.1.1. Frequency of Species-Specific Markers in Hybrids of *S.* subgen. *Vetrix*

Hybrids in the CCV dataset are cultivated hybrids that share morphological characters with the expected parental species (*Salix caprea*, *S. cinerea,* and *S. viminalis*). The proportions of SSM specific to *S. caprea*, *S. cinerea,* or *S. viminalis* in expected hybrids were assessed and compared using different counting approaches. Considering all SSM together with unassigned markers, the proportion of markers in hybrids is too biased to provide a clear insight into the origin of hybrids (Figure 1a). Further filtering of markers (present in at least one half of the samples of the species) resulted in better resolution (Figure 1a–d). Only eight samples were consistently identified as hybrids of the studied species: six samples as *S. caprea* × *S. viminalis* (=*S.* ×*smithiana* Willd.), one sample as *S. cinerea* × *S. viminalis* (*S.* ×*holosericea* Willd., syn.: *S.* ×*dasyclados* Wimm.), and one cultivated sample was undoubtedly identified as an extreme morphotype of *S.* ×*rubens* Schrank. (*S. alba* × *S. fragilis*) (Figure 1b,d). Three samples (G35, G42 and G43) probably represent pure species (G35 *S. cinerea* and G43 *S. viminalis*), G42 consists mainly of SSM of *S. caprea* (Figure 1d). The remaining hybrids were not unambiguously resolved, all of them containing a considerable part of the genome of one of the three expected species (Figure 1c), but apparently containing unidentified markers (Figure 1b,d), either representing markers of the three species studied but not collected in our sampling, or markers of other species. In particular, samples G33, G34, G20, G24, G48, and G49 had a high proportion of unidentified markers (Figure 1d).

Comparison of expected hybrids with other species of the subgenus *Vetrix* revealed the presence of SSM from *S. purpurea* in samples G48 and G49 (Figure 2), suggesting their hybrid origin from *S. purpurea* × *S. viminalis* (*S.* ×*rubra* Huds.). Other compared species (*S. aurita* L., *S. eleagnos* Scop., *S. hastata* L., *S. myrtilloides* L., *S. rosmarinifolia* L., and *S. silesiaca* Willd.) did not have their SSM in the tested hybrids and therefore the data are not presented here. Additional samples of naturally occurring hybrids of *S. caprea* × *S. viminalis* and *S. cinerea* × *S. viminalis* were extracted from the ‘*has-myr*’ dataset and the patterns of the proportion of SSM are comparable to the profiles of the cultivated plants, i.e., samples G70–G76 are similar to samples from the wild (X04 and X24) and sample G37 is similar to X06 (Figure 2).

#### 2.1.2. Frequency of Species-Specific Markers in Hybrids of *S.* subgen. *Salix*

The higher number of species-specific markers of taxa in the subgenus *Salix* (Table 1) provided better resolution when comparing hybrids. Both cultivated (experimental) and natural hybrids were mostly identified as *S. alba* × *S. fragilis* (*S.* ×*rubens*) hybrids with about one half of the markers of each parent (Figure 3). Eleven samples had approximately only one quarter of the *S. alba* SSM, indicating possible backcrossing with *S. fragilis*. One sample, originally grown as *S.* ×*rubens*, was identified as a *S. fragilis* × *S. pentandra* hybrid. Samples A04 and A08 are cultivated trees from public greenery with distinct weeping branches, but based on the proportion of SSM they represent a form of *S. alba*. In contrast to the CCV dataset, the proportion of unidentified markers is quite low among the samples (Figure 3c); relatively higher proportions were observed only in samples of *S. triandra* (A106) and *S. excelsa* S.G.Gmel. (A107), both from Transcaucasia. A small proportion of unidentified SSM is present in samples of ornamental cultivars of *S. babylonica* L. hybrids (A03, A07, A72).

Several naturally occurring samples were initially expected to be hybrids with *S. triandra* (see Appendix A). Most of them were identified as hybrids of *S. alba* × *S. fragilis* (e.g., A48), or extreme morphotypes of *S. alba* (A21) or *S. fragilis* (e.g., A55).

### 2.2. Detection of Hybrids Using Binary SNP Data

Cluster analysis of both datasets showed a clear separation of the taxa. Despite the lower number of SSM for the *Vetrix* species, clustering analyses (PCoA; Figure 4) and the calculation of the neighbour-joining dendrogram confirmed that the species are well separated (Figure 5). Principal coordinate analysis confirmed an intermediate position between the parents for the hybrids *S. caprea* × *S. viminalis* and *S. cinerea* × *S. viminalis*, invariably in all three major axes of variability (Figure 4). Similar intermediate positions of these hybrids were calculated by SplitsTree (Figure 5). The hybrids of unknown origin identified by scoring of SSM (Figure 1d) show some intermediate position (G33, G34, G24) and have variable positions that differ among axes, confirming the expectation that they represent hybrids of studied taxa with another species. Sample G78 has a stable position in all axes, suggesting its putative backcross origin (Figure 4). Hybrid *Salix purpurea* × *S. viminalis*, identified by SSM scoring (Figure 2), has a false intermediate position between *S. cinerea* and *S. viminalis* on two major axes, but is clearly separated from this species on the third axis, clearly showing that it is another taxon (Figure 4). Neighbor-joining by SplitsTree could not identify these samples as different and incorrectly placed them between the analysed taxa. Species of the subgenus *Salix* are even more separated by both cluster analyses (Figure 6, Figure 7, and Appendix A), *S. triandra* appears to be distant from species of the subgenus *Salix* (Figure 7 and Appendix A). PCoA revealed two distinct clusters of *S. alba* × *S. fragilis* hybrids, supporting the idea that these hybrids are repeatedly backcrossed with *S. fragilis* (Figure 6). Putative *S. fragilis* × *S. pentandra* hybrids are intermediate between the parents on all three major axes. The Armenian sample of *S. triandra* clusters with European samples of the species in both the PCoA (Figure 6) and the neighbour-joining dendrogram by SplitsTree (Figure 7 and Appendix A), whereas *S. excelsa* differs from *S. alba* in both cluster analyses. The ornamental cultivars of *S. babylonica* hybrids also cluster relatively close to *S. babylonica* ‘Babylon’ in both analyses.

### 2.3. Inferring Ancestry by Structure

Bayesian analysis by Structure works well for species of subgenus *Salix*. All natural hybrids were confirmed, backcrosses were clearly identified, all hybrids were revealed by SSM scoring (Figure 3), and clustering analyses (Figure 6 and Figure 7) were also confirmed by Structure (Figure 8b). However, two taxa appear to be biased: sample A107 (*S. excelsa*) was identified as *Salix alba*, which also contains all other species clusters. This result is not fully consistent with other analyses; the sample is closely related to but distinct from *S. alba* in other analyses. Second, controversial are the results for *S. babylonica* hybrids (A03, A07 and A72), which were not recognised as hybrids by the Structure. In the *Vetrix* species, the doubtful taxa are identified by structure as hybrids of the analysed taxa, with the exception of G53 and G34 (Figure 8a). 

## 3. Discussion

### 3.1. Differentiating Willow Species Using DArTseq Markers

The high potential of DArTseq markers for species identification was demonstrated by the analysis of 53 samples of four willow species from Poland [19]. The large number of markers potentially provides a robust tool for hybrid identification. However, only *S. purpurea* and *S. viminalis* hybridised quite frequently; other species from the study [19] either hybridise rarely (*S. triandra* × *S. viminalis*), or their existence has not been reliably documented. *Salix alba* and *S. viminalis* are known for their high hybridisation potential [22,24], and our study confirms that additional related taxa that are putatively hybridising with the studied taxa—namely *S. fragilis*, *S. caprea* and *S. cinerea*—are genetically well separated (Table 1, Figure 4, Figure 5, Figure 6 and Figure 7). The number of species-specific markers is comparable to that of [19], suggesting broad applicability of DArTseq markers to different willow species. Although for other species—*S. triandra*, *S. pentandra*, *S. excelsa*, *S. babylonica,* and *S. purpurea*—only single or few samples were tested, we confirmed that these taxa are also well separated genetically. Therefore, the DArTseq markers have potential for hybrid identification in other problematic hybrid groups such as weeping willows or for resolving the *S. alba*–*S. excelsa* controversy. The single sample of *S. excelsa* has the closest position to *S. alba* of all the other taxa (Figure 7 and Figure 8), but it differs significantly (Figure 6). All analyses together identified the most divergent species as *S. triandra*, which differs significantly from both groups, the species of *S.* subgen. *Salix* and *S.* subgen. *Vetrix* (Appendix A).

### 3.2. Using DArTseq Markers to Identify Willow Hybrids

Three types of willow hybrids occur frequently in our samples—*S. alba* × *S. fragilis*, *S. caprea* × *S. viminalis,* and *S. cinerea* × *S. viminalis* (see also Appendix A), i.e., the hybrid combinations that were confirmed. However, we also identified two other (unexpected) hybrids among the cultivated samples from the experimental willow garden for biomass production: *S. fragilis* × *S. pentandra* and *S. purpurea* × *S. viminalis*. 

From a technical point of view, DArTseq markers serve as a very powerful tool to discriminate morphologically similar hybrids, especially useful for *S. caprea* × *S. viminalis* and *S. cinerea* × *S. viminalis* hybrids, i.e., an important source for the production of novel crops for biomass production. In the *CCV* dataset, few samples resembled the multi-hybrid profile (Figure 1 and Figure 8a), but additional comparison with another *Vetrix* species, *S. purpurea*, confirmed that two samples were undoubtedly hybrids, *S. purpurea* × *S. viminalis*. This is an important issue, because a hybrid combination of different parentage than that tested (i.e., missing parent for comparison) is not erroneously assigned to any of the pure species tested but is recognisable by the unusual pattern. The hybrid of unknown origin must be further compared with additional samples of other species to find the unknown parent. 

### 3.3. Asymetric Introgression in S. alba × S. fragilis Hybrids

Although this study was not designed to describe biological processes, some interesting conclusions can be drawn about hybridisation patterns. The most striking result is the pattern of hybridisation between *S. alba* and *S. fragilis*. The DArTseq markers successfully identified the expected and unknown samples of *S. alba* × *S. fragilis* F_1_ hybrids, with all tested approaches clearly separating the F_1_ hybrid from its parents in both natural and cultivated samples. Several hybrids (all of natural occurrence) were identified as backcross hybrids with *S. fragilis*, making a superficial genetic and morphological gradient between the F_1_ hybrid and *S. fragilis* (see also Appendix A for their morphology). On the contrary, no backcross hybrids with *S. alba* were found in our sample, suggesting that asymmetric introgression between *S. alba* and *S. fragilis* occurs in the study area. Backcross hybridization is already documented in willows, in populations of *S. helvetica* Vill. × *S. purpurea* on glacial forefield in the Alps, and backcrossing to both parents was observed without significant deviations of RADseq loci from expected segregation patterns [16]. However, in the *S. alba* × *S. fragilis* hybrid population, located in the well-preserved alluvial forest of the Morava river, backcrossing with one of the parents—*S. alba*—is completely absent. Whether the absence of backcross hybrids with *S. alba* is caused by local ecological factors (natural selection) or by other biological factors needs further investigation. In Latvia, at the edge of the distribution of *S. alba*, hybridisation is predominantly unidirectional, by fertilisation of *S. fragilis* with pollen from *S. alba*. Extended sampling across Europe will shed light on whether asymmetric introgression in *S. alba* × *S. fragilis* hybrids is a general or local phenomenon. However, even if it is only a local phenomenon, at the contact zone between flat and hilly regions, it could explain the false impression that in Europe only *S. alba* and its hybrid occur [26] (p. 73), and that *S. fragilis* is ‘dissolved’ in the hybrid.

### 3.4. Hybrids of S. viminalis

Hybrids of *S. viminalis* with *S. caprea* and *S. cinerea* are morphologically (to the untrained eye) easily interchangeable, but the patterns of hybridisation differ. The homoploid hybrid *S. caprea* × *S. viminalis* forms a rather uniform cluster, whereas the heteroploid hybrids of *S. cinerea* × *S. viminalis* are represented by genetically diverse accessions. Backcrossing could be the explanation, as well as unequal proportions of parental genomes due to different ploidy of the parents (*S. viminalis* diploid, *S. cinerea* tetraploid). We must add that our results could be biased by the origin of the plants: although they are hybrids of spontaneous origin, they are derived from cultivated material and therefore do not represent plants growing under natural conditions.

### 3.5. Taxonomic and Systematic Implications

*Salix triandra* has traditionally been placed within the subgenus *Salix* (consisting of late-flowering, narrow-leaved taxa; [26,27], but it differs significantly from other taxa in the subgenus, occupying an intermediate position between the subgenera *Salix* and *Vetrix* (Figure 7 and Appendix A). Our data provide further confirmation to previous observations of clear genetic differentiation of *S. triandra* from the subgenus *Salix* [28,29,30] and support the redefinition of the species position in willow systematics.

*Salix excelsa* is a widespread species of western and central Asia, extending into semi-arid regions. It’s relationship to *Salix alba* has long been debated [26,31], and morphological intermediates are known throughout the range of the species. Although our sample was analysed from a single individual, which does not allow us to draw any serious conclusions, it shows that the species is genetically very closely related to *S. alba*, sharing more than half of the genome with the European and West Asian samples of *S. alba*. The species appears to be of hybridogenous origin, but the variation and genomic composition of the species is still under investigation.

The nomenclature of *Salix fragilis*—the crack willow—underwent a revolutionary change in the last decade. Based on Skvortsov’s expectation [26] (p. 73) that *S. fragilis* was either rare or absent in Europe and that all so-called crack willows were in fact hybrids, the new name *S. euxina* was proposed for the ‘pure’ crack willow and the vast majority of European crack willows were arbitrarily considered as hybrids [32]. However, this change was based solely on morphological studies and lacked a solid background of genetic analysis. The risk of misidentification has recently been documented [33]; the number of ovules in Gynaecea does not match Belyaeva’s expectations.

Our data confirm the frequent hybridisation of *S. fragilis* with *S. alba*, and we also showed the tendency for unidirectional backcrossing, which—at least locally—predominates and makes the morphological differentiation between the backcross hybrids and the ‘pure’ crack willow difficult (cf. Appendix A). This makes the identification of taxa on herbarium specimens even more difficult. Considering that herbarium specimens often consist of small branch fragments, not always with well-developed leaves or other characteristics, one could easily misidentify the hybrid from the parent species. This could give the false impression that crack willow exists only as a hybrid—the hybrid represents a morphological continuum to the parent species and is frequently documented in herbarium collections. However, some morphological characters, such as the presence/absence of trichomes or the number of ovules [31], help to clearly distinguish hybrids from the parent species. We have confirmed that crack willow (*S. fragilis*) as a ‘pure’ species is common in Europe and is an important part of alluvial habitats in the hilly and submontane regions of Central Europe.

Obviously, not all taxa resembling crack willow described from Europe can be automatically considered synonymous with ‘hybrid’ crack willow. Our data confirm that ‘pure’ crack willow is morphologically highly variable but clearly distinct from hybrids with *S. alba*. Several published names undoubtedly refer to ‘pure’ willow—such as *Salix fragilissima* Host [34] (compare p. 6–7, Figures 22 and 23 in [34] with Appendix A in this study)—which have nomenclatural priority over the name *S. euxina*. However, our results support the opinion of [33] for the re-acceptance of the classical name *S. fragilis* for the pure species. The neotype for *S. fragilis* therefore must be established on the basis of genetic data and, ideally, be confirmed by the number of ovules and other morphological characters.

In our study, we tested plants resembling (by leaf morphology) the diploid species *S. triandra*, including morphotypes similar to plants on Tausch’s original *S. alopecuriodes* material (expected to be hybrids of *S. fragilis* × *S. triandra*), or Host’s *Salix speciosa* [34] (Figure 17). Although high numbers of *S. alba* × *S. fragilis* hybrids (including backcross hybrids) were documented and a single *S. fragilis* × *S. pentandra* hybrid was confirmed, none of the hybrids between tetraploids (*S. alba*, *S. fragilis, S. pentandra*) with diploid *S. triandra* were confirmed. All samples suggesting potential hybridisation with *S. triandra* were confirmed to be either *S. alba* × *S. fragilis* (the vast majority) or, in particular, the backcross hybrids with *S. fragilis*, including the Tausch’s morphotype *S. alopecuroides* or Host’s *S. speciosa* (see Appendix A). The backcross hybrids show enormous phenotypic variation, which could lead to misidentification. This raises the burning question of whether the hybrids between the tetraploid species and the diploid *S. triandra* really exist, and whether they are (as is apparently the case with Tausch’s *S. alopecuroides* and Host’s *S. speciosa*) only extreme morphological deviations of different hybrid crosses.

## 4. Materials and Methods

### 4.1. Plant Material

We studied 179 willow samples, consisting of 10 species and their hybrids. The samples were analysed separately as two datasets: The *CCV* dataset contains taxa of *Salix* subgen. *Vetrix* (3 species, project DWl15-1960), the *AF* dataset contains taxa of *Salix* subgen. *Salix* (6 species, project DWl19-4801). In addition, the *CCV* dataset was compared with 5 samples of *S. purpurea* (*has-myr* dataset, project DWl20-5556, not presented in this study). All samples collected by RJV in the field are documented by the herbarium voucher specimen deposited in OL (Botany Department of Palacký University in Olomouc). Samples provided by JW were cultivated in the experimental garden of the Silva Tarouca Research Institute for Landscape and Ornamental Gardening in Průhonice. Taxonomic information and the geographical origin of the studied plant material are summarised in Appendix A, with images of selected samples (Appendix A).

#### 4.1.1. *CCV* Dataset

The *CCV* dataset consists of 77 samples of *Salix* subgen. *Vetrix*, namely 3 species: *S. caprea*, *S. cinerea,* and *S. viminalis*, and their (expected) hybrids *S.* ×*smithiana* (*S. caprea* × *S. viminalis*), *S.* ×*holosericea* (*S. cinerea* × *S. viminalis*). In our study, the hybrid *S.* ×*holosericea* is identical to the taxon traditionally named *S.* ×*dasyclados* Wimm. in Europe but is not conspecific with the Siberian taxon *S. gmelinii* Pall. All samples originate from the collection of JW. For further identification of unknown hybrids, 5 samples of *S. purpurea* and 3 additional hybrids of *S. viminalis* were co-analysed from other unpublished project (*has*-*myr*, Vašut et al. in prep., Appendix A).

#### 4.1.2. *AF* Dataset

The *AF* dataset consists of 94 samples of *Salix* subgen. *Salix* from Europe (and a few from western Asia), with the vast majority of samples originating from Czech Republic; the large sample originates from riparian forests along the Morava river in central Moravia. About ¼ of the samples (22) represent genotypes for and from experimental crosses (provided by JW), selected with the aim of identifying the parents in putative hybrids. Another part (70 samples) consists of natural taxa sampled in the field (by RJV and JL), both pure species (*S. alba*, *S. fragilis* and additional taxa for comparison—*S. pentandra*, *S. triandra*, *S. excelsa,* and *S. babylonica*) and natural hybrids of different morphotypes, with the aim of delimiting the genetic variation of pure species and comparing natural hybrids with artificial ones. Seven samples of ornamental willows were grown in urban green spaces. In our study, the crack willow is named as *Salix fragilis* L., following recent nomenclatorial conclusions [33].

### 4.2. DNA Isolation and DArTseq Sequencing

Total genomic DNA was extracted according to the Plant DNA Extraction Protocol for DArT [35], mostly from fresh leaf tissue and some samples from dried leaf tissue (see Appendix A). DNA concentration was measured using a Nanodrop, inspected on a 0, 8.0% agarose gel and diluted to the required concentration: 20–30 µL restriction enzyme digestible genomic DNA, free of endonucleases, dissolved in EB buffer (10 mM Tris-Cl, pH 8.5) at a concentration of 50–100 ng/µL. Extracted genomic DNA samples were sent to Diversity Arrays Technology Pty Ltd. (Canberra, Australia) (http://www.diversityarrays.com, accessed on 1 July 2019) for the *Willow DArTseq (1)* whole genome genotyping service, with the procedure described in [17,18,19].

DArTseq results are presented in two types of datasets: (i) co-dominant single nucleotide polymorphism (SNP) markers indicating the SNP in heterozygous state (1), the SNP in homozygous state (2) and no mutation in the sequence (0); and (ii) dominant SilicoDArT markers indicating the presence (1) or absence (0) of restriction fragments. 

The marker sequences of co-analysed datasets *AF*+*CCV* were aligned to the reference genome of *Populus trichocarpa* genome [36].

### 4.3. Frequency of the Species-Specific Markers

The identification of ‘species-specific’ markers followed the approach used for the detection of *Festuca* × *Lolium* hybrids [37]. We identified species-specific markers (SSM) based on the results of SilicoDArT data. The marker was considered species-specific if it had a positive score (1) in at least one sample of the species, but a negative score (0) in all samples of other species (‘All SSM’). We also considered only those markers that had a positive score (1) in at least half of the samples of the species, but a negative score (0) in all samples of other species (‘SSM > 50%’). In order to overcome the bias caused by the unequal number of samples per species, we normalised the value of ‘SSM > 50%’ to the number of all SSM per species and to the average number of SSM per sample of the species.

### 4.4. Statistical Analyses

Genetic differentiation of samples from each dataset was analysed by principal coordinate analysis (PCoA), calculated and visualised in R using the dartR [38,39], ggplot2 [40], and adegenet [41] libraries. The data matrix was filtered to exclude monomorphic loci, resulting in 85,915 binary SNPs for the CCV dataset and 154,752 binary SNPs for the *AF* dataset.

Species relationships were investigated by constructing the phylogenetic network in SplitsTree 5 [42]. The input file was generated in R using the dartR library, based on the same data matrices used in the PCoA. We ran the computations using the default settings, i.e., implementing uncorrected P/Hamming distances (distance matrices), NeighborNet (splits), and the equal angle convex hull algorithm (splits networks).

To infer the ancestry of putative hybrids in each dataset, we used the Bayesian clustering approach implemented in the program Structure (ver. 2.3.4.) [43]. The input data matrix was generated in R using the dartR library (function gl2structure) [39]. The analysis was performed with an admixture model with K between 1 and 10 with 10 replicate runs for each K, 10,000 burn-in iterations followed by 100,000 MCMC iterations. The ΔK statistics [44] was used to find the appropriate number of clusters in the Structure output files, using the Structure Harvester [45]. The best fitting K was recalculated using 10 replications with 50,000 burn-in iterations followed by 500,000 MCMC iterations. The results were inspected using the Clumpak [46], and the final bar plots were manually edited accordingly (reordering samples and recolouring bars).

## 5. Conclusions

We demonstrated that genome-wide DArT-seq sequencing facilitates the reliable identification of willow hybrids as an alternative to RAD-seq. Although our study was designed to test the applicability of the method, we have already demonstrated that it provides new insights into evolutionary processes in willows and important knowledge for the systematics of the genus. We recommend an analysis pipeline that combines both dominant SilicoDArT data for quick overview and co-dominant SNP data for detailed but computationally expensive analysis. The method has applications in biotechnology, horticulture, ecology, nature protection, and other fields. 

## Figures and Tables

**Figure 1 plants-13-00639-f001:**
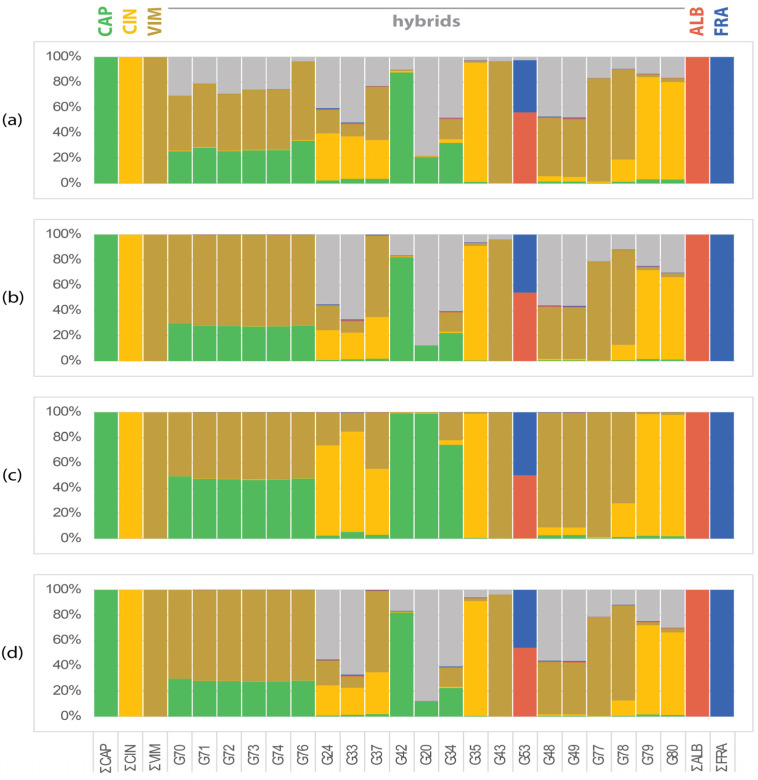
Frequency of species-specific markers in the *Vetrix* hybrids (*CCV* dataset). (**a**) All SSM: proportion of all detected species-specific markers present at least in 1 sample of the species; (**b**) SSM > 50%: proportion of SSM present in >50% of samples of the species; (**c**) proportion of SSM present in >50% of samples of the species proportioned to average number of SSM per sample of the species; (**d**) proportion of SSM present in >50% of samples of the species proportioned to total number of SSM of the species. Samples G37 and G78 were identified as hybrids *S. cinerea* × *S. viminalis*, samples G71–G76 as *S. caprea* × *S. viminalis*. Samples G48 and G49 appeared to be hybrids of *S. viminalis* with other species, Figure 2 shows that unknown parent is *S. purpurea*. Samples G24, G33, G42, G20, G79, and G80 remain unidentified, representing hybrids of *S. caprea* (G20 and G42), *S. viminalis* (G77), *S. cinerea* (G79 and G80), or *S.* ×*holosericea* (syn. *S.* ×*dasyclados*) with other unidentified species. Sample G53 was identified as *S*. ×*rubens*. Legend to taxa: CAP = *S. caprea*; CIN = *S. cinerea*; VIM = *S. viminalis*; ALB = *S. alba*; FRA = *S. fragilis*; none = markers present in hybrids only not belonging to any species of the *CCV* or *AF* dataset.

**Figure 2 plants-13-00639-f002:**
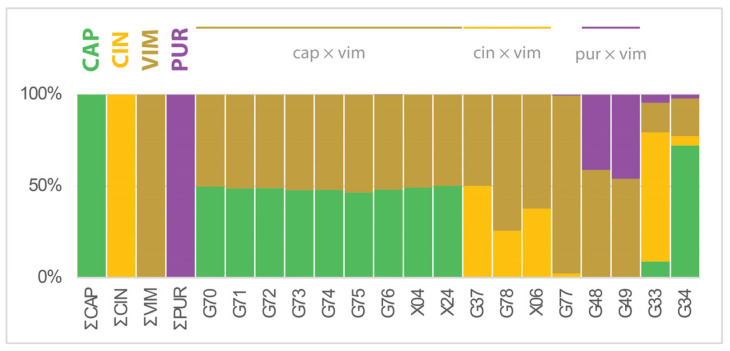
Frequency of species-specific SSM in hybrids (*Salix* subgen. *Vetrix*) compared with *S. purpurea*. Thirteen hybrids from the CCV dataset of uncertain origin were additionally tested for the presence of species-specific markers from seven other *Vetrix* species (*has-myr* dataset). Two samples were identified as *S. purpurea* × *S. viminalis* hybrids, 3 hybrid samples (from the *has-myr* dataset) collected in the wild (2 samples of *S. caprea* × *S. viminalis*, 1 sample of *S. cinerea* × *S. viminalis*) have the same profiles as hybrids from the experimental garden (CCV dataset). A further 11 hybrids of uncertain origin were not positive for the presence of markers of *S. aurita*, *S. hastata*, *S. myrtilloides*, *S. rosmarinifolia,* and *S. eleagnos*; presented are samples G33 and G34 showing uncertain pattern. The graph shows proportion of SSM presented in >50% of samples with values recalculated to average number of SSM per sample—compare with Figure 1c. CAP = *S. caprea*; CIN = *S. cinerea*; VIM = *S. viminalis*; PUR = *S. purpurea*.

**Figure 3 plants-13-00639-f003:**
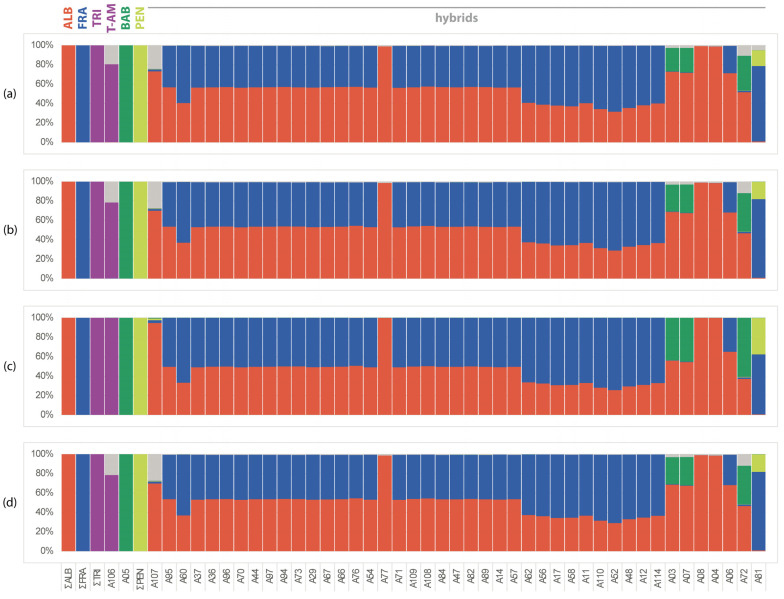
Frequency of species-specific markers in *Salix* hybrids (*AF* dataset, *Salix* subgen. *Salix*). (**a**) All SSM: proportion of all detected species-specific markers present at least in 1 sample of the species; (**b**) SSM > 50%: proportion of SSM present in >50% of samples of the species; (**c**) proportion of SSM present in >50% of samples of the species proportioned to average number of SSM per sample of the species; (**d**) proportion of SSM present in >50% of samples of the species proportioned to total number of SSM of the species. ALB = *S. alba*; FRA = *S. fragilis*; TRI = *S. triandra*; PEN = *S. pentandra*; BAB = *S. babylonica* ‘Babylon’; none = markers present in hybrids only not belonging to any species of the *CCV* or *AF* dataset; T-AM = *S. triandra* from Transcaucasia (Armenia).

**Figure 4 plants-13-00639-f004:**
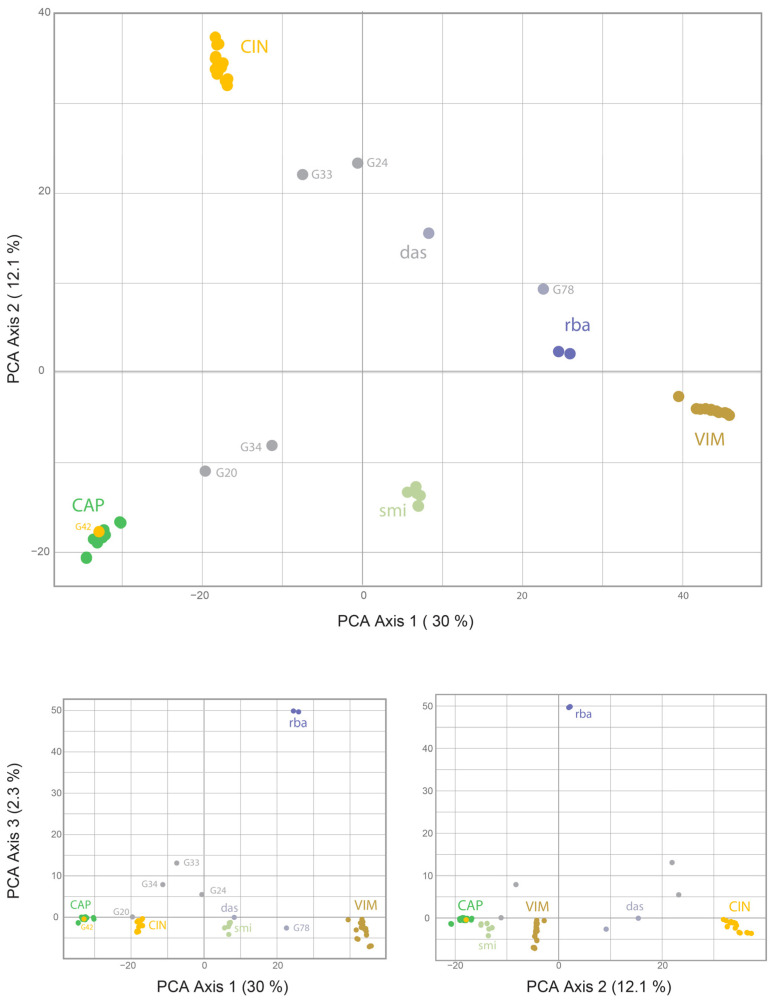
Principal coordinate analysis of *Salix* hybrids (*CCV* dataset, *Salix* subgen. *Vetrix*) based on 85,915 binary SNPs of 77 samples, showing 3 major axes of variability. CAP = *S. caprea*; CIN = *S. cinerea*; VIM = *S. viminalis*; smi = *S. caprea* × *S. viminalis*; das = *S. cinerea* × *S. viminalis*; rba = *S. purpurea* × *S. viminalis*.

**Figure 5 plants-13-00639-f005:**
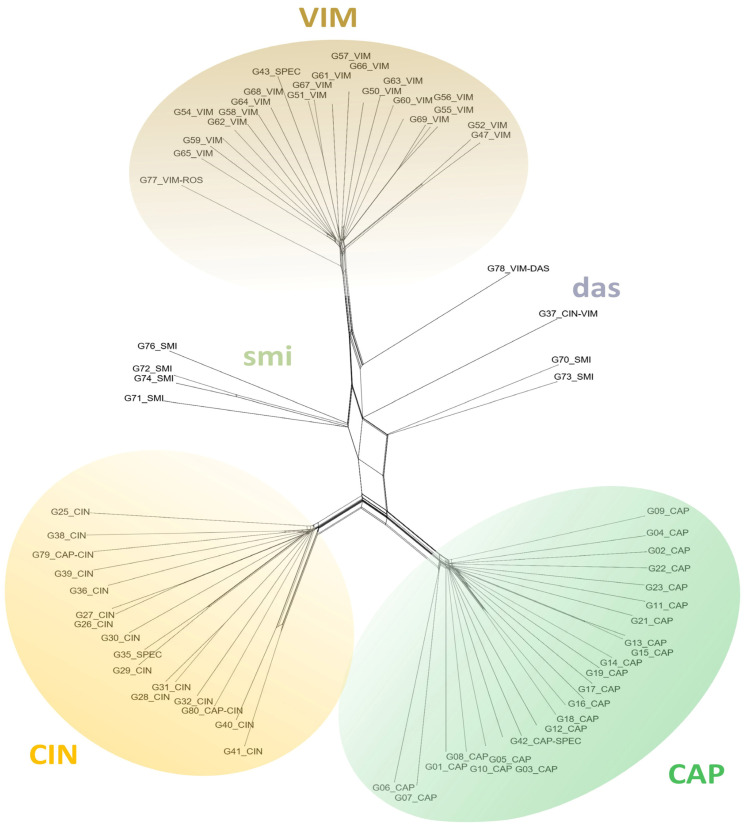
Neighbour-joining unrooted dendrogram for *Salix* subgen. *Vetrix* based on 85,915 binary SNPs generated in SplitsTree (Fit: 99.94). CAP = *S. caprea*; CIN = *S. cinerea*; VIM = *S. viminalis*; smi = *S. caprea* × *S. viminalis* (*S.* ×*smithiana*); das = *S. cinerea* × *S. viminalis.* Parental taxa (*S. caprea*, *S. cinerea*, *S. viminalis*) are well separated, hybrids *S. caprea* × *S. viminalis* and *S. cinerea* × *S. viminalis* have intermediate position, however, hybrids of different origin share the false intermediate position.

**Figure 6 plants-13-00639-f006:**
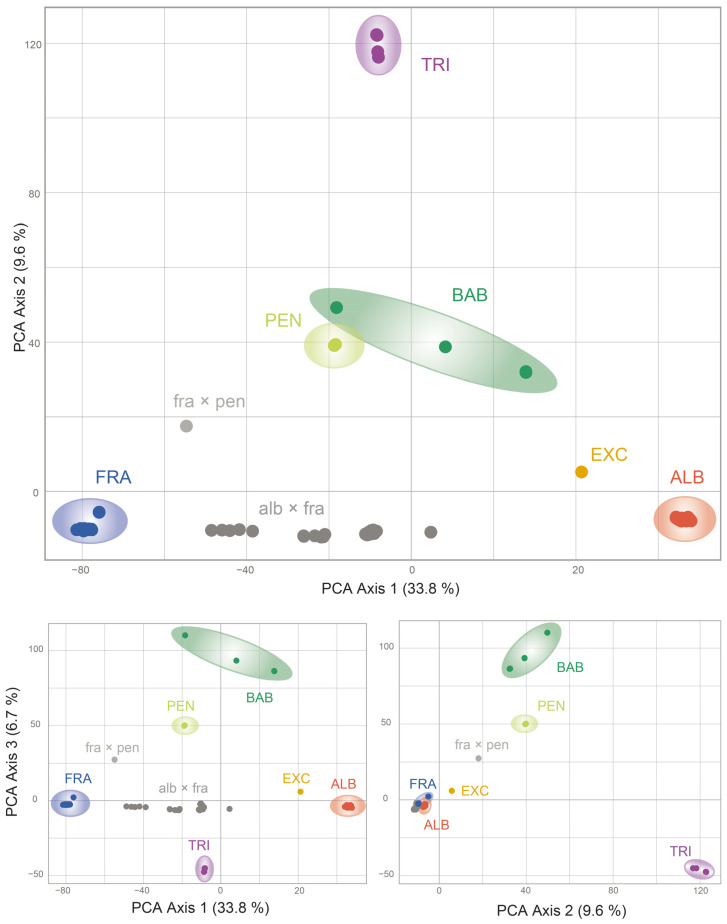
Principal coordinate analysis of *Salix* hybrids (*AF* dataset, *Salix* subgen. *Salix*) based on 154,752 binary SNPs of 94 samples, showing 3 major axes of variability. ALB = *S. alba*; FRA = *S. fragilis*; TRI = *S. triandra*; PEN = *S. pentandra*; BAB = *S. babylonica* s.l., EXC = *S. excelsa*.

**Figure 7 plants-13-00639-f007:**
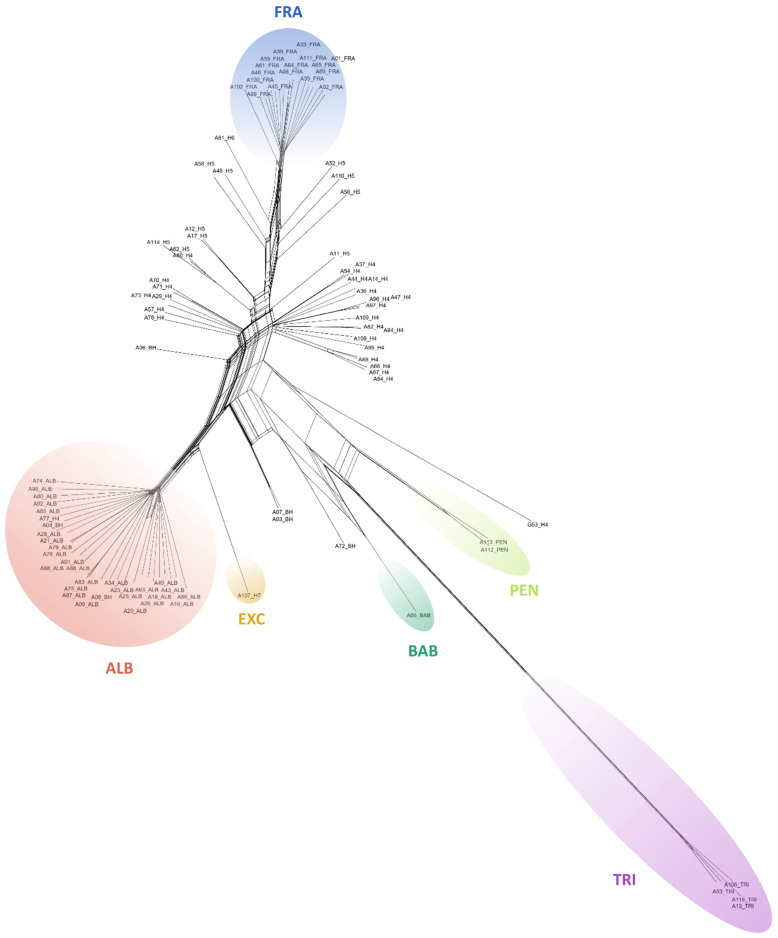
Neighbor-joining unrooted dendrogram for *Salix* subgen. *Salix* based on 154,908 binary SNPs generated in SplitsTree (Fit: 99.20). ALB = *S. alba*; FRA = *S. fragilis*; TRI = *S. triandra*; PEN = *S. pentandra*; BAB = *S. babylonica* ‘Babylon’, EXC = *S. excelsa.* All taxa are well separated, most distinctly *S. triandra*. Hybrid *S. alba* × *S. fragilis* forms distinct cluster in the middle part, backcross hybrids with *S. fragilis* appear half-way between the F_1_ hybrid and FRA. Other hybrids also have an intermediate position between their parents.

**Figure 8 plants-13-00639-f008:**
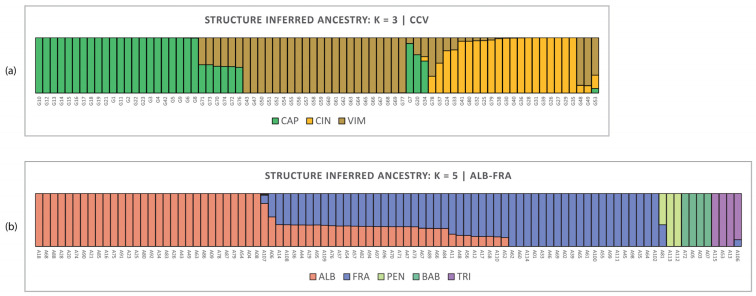
Inferred ancestry by the Structure analysis. Bar plots of individual assignment probabilities for (**a**) *CCV dataset*, *Salix subgen. Vetrix*; *K* = 3, CAP = *S. caprea*; CIN = *S. cinerea*; VIM = *S. viminalis***;** (**b**) *AF* dataset, *Salix* subgen. *Salix*; *K* = 5, ALB = *S. alba*; FRA = *S. fragilis*; TRI = *S. triandra*; PEN = *S. pentandra*; BAB = *S. babylonica* ‘Babylon’.

**Table 1 plants-13-00639-t001:** Species-specific markers. Total number of species-specific markers (SSM) based in SilicoDArT markers: All SSM counts markers present only in single species (at least single one), SSM > 50% counts species-specific markers, which are present in at least 50% of samples of the species. Σ summarize all observed SSM in all samples, Ø counts average number of SSM per sample of the species.

Species	All SSM		SSM > 50%	
	Σ	Ø	Σ	Ø
*S. caprea*	28,598	4161	2365	1781
*S. cinerea*	19,117	4380	2529	1810
*S. viminalis*	14,546	5643	5263	4126
*S. triandra*	36,107	25,117	24,460	21,234
*S. alba*	50,917	25,803	24,525	20,438
*S. fragilis*	25,420	18,488	20,566	17,283
*S. pentandra*	7559	6376	7559	6376
*S. babylonica* ^1^	10,676	(10,676)	(10,676)	(10,676)

^1^ Counted for single sample of *S. babylonica* ‘Babylon’.

## Data Availability

Data is contained within the article and Appendix A. Additional information is available on request from the corresponding author.

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
