# Peer review of "Detection of Hybrids in Willows (Salix, Salicaceae) Using Genome-Wide DArTseq Markers"

_plants, 2024, doi:10.3390/plants13050639_

Round 1

Reviewer 1 Report

Comments and Suggestions for Authors

General Comments:

I have reviewed the manuscript “Detection of hybrids in willows (Salix, Salicaceae) using genome wide DArTseq markers” and find it to be acceptable for publication in its present form with only minor revisions. The information in the manuscript is original, and the topic is of considerable interest suitable for publication in the Plants.  

The taxonomic difficulties of the species complex “Salix alba, S. fragilis, their hybrids”, as well as some taxa in the subgenus Vetrix, were emphasized by many previous publications. The discrimination of  Salix alba, S. fragilis and their hybrids is still not fully resolved and a clarification of their taxonomy is an important and valuable attempt. The authors demonstrated that genome-wide DArT-seq sequencing assists in the reliable identification of willow hybrids and provides insights into taxonomy and systematics of these taxa. Interestingly, the authors discovered the phenomenon of the asymmetric introgression in S. alba × S. fragilis hybrids.  There are valuable conclusions about the genetic differentiation of S. triandra. Also, the authors provided interesting insights about a controversial species S. euxina, which was proposed by Belyaeva (2009) because of a lack of appropriate type material for S. fragilis. The molecular data suggested that the pure S. fragilis occurs naturally in Europe.  The authors recommend to continue to use the name "S. fragilis" for a glabrous crack willow, and I agree with the authors that the original Linnean names should be maintained. Finally, the authors suggest that an appropriate neotype for S. fragilis should be established based on the molecular data, which is an extremely valuable proposal.

Specific comments:

Throughout the paper the authors used the present and past tenses interchangeably. Suggest using only the past tense.

For example (Line 138) “In particular, samples G33, G34, G20, G24, G48 and G49 have a 138 high proportion of unidentified markers (Figure 1d).” Replace “have” with “had’.

The species names should have the authors when they first mentioned.

The authors used the term “biotopes”. In English-speaking countries, the term “habitat” is used more frequently. It is up to the editors which term is better to use in this publication.

The publication needs some minor revisions:

Line 18-18 is missing a verb: “A total of 177 willow samples from six species of Salix subgen. Salix and four 18 species of Salix subgen. Vetrix, including those used in biomass crop production, representing important European taxa.”

Lin 25 delete or replace “However”, it is out of place here.

Line 41 delete “the” before “carbon”

Line 52 delete “etc.” 

Line 65 delete ”species affiliation for important accessions and so forth”

Line 67 replace “vegetation science” with “plant science”

Line 69 replace  “accurate” with “various”

Line 71 I am not sure that “willows have reduced flowers (and thus lack distinct characters)” is an accurate statement

Line 72 Revise “… requires repeated visits to the population to understand the relationship between flowers, fruits and leaves”

Line 97 eliminate the used of one “aim” in “We aim to use this method in experimental crosses and breeding aimed to “

Line 168 Add “The” before “Graph show

 Line 175 replace “Have” with “had’

Line 178 delete “agg” in “S. babylonica agg”  

Line 184 clarify “originally thought to be” in “occurring samples were originally thought to be hybrid”

Lin 220 what dies “s. str.” means  in “S. babylonica s. str.”

Line 237 “das” should not be bold

 Line 239. Comma after however

Line 271 replace “in their study” by the author names

Line 290 delete “expected to be”

Line 298 Replace “comparison with other “ with “comparisons with another”

Line 305 clarify “was not designed to describe biological processes”

Line 312 delete “for their morphology”

Line 315 replace “in population” with “in populations”

Line 335 what is “spontaneously generated’?

Line 343 replace “subgenera” with the “subgenus”

Line 348 delete or re-phrase “which does not allow any serious conclusions to be drawn”,

Line 359 delete “by”       

Line 364-366 I am not sure if this is accurate: “Considering that herbarium specimens often consist of small branch fragments, not always with well-developed leaves or other characteristics, one could get the impression that crack willow might only exist as a hybrid. “ Needs revision

Line 414 delete “all over”

Lilne 424 replace “following nomenclatorial conclusions of [33]” with following recent nomenclatorial conclusions [33]”

Comments on the Quality of English Language

only minor edits are required - please see above

Author Response

I am grateful to the reviewer for useful comments on our manuscript. All comments have been accepted and the text has been revised accordingly. None of the suggested corrections caused controversy and all improved our manuscript. I am glad that the reviewer supports the idea of establishing a neotype for Salix fragilis, but this should be done later, ideally in collaboration with authors who have previously proposed a similar idea. 

Reviewer 2 Report

Comments and Suggestions for Authors

The article is quite interesting and the methodology adopted appears appropriate. I have marked in the file some parts of the M&M and of the results that were not clear when reading. On the whole the Ms deserves to be published on Plants.

Comments on the Quality of English Language

In my opinion the Language used is clear and correct.

Author Response

We thank the reviewer for useful comments. 
1) The abstract has been corrected according to the reviewer's recommendation.

2) Note on line 269-270 is unclear to us (RJV & JW). The majority of the dataset needs to contain the parental taxa to provide a solid data base for species delimitation, hence the majority of the samples. In the CCV dataset we emphasise the clear hybrids (S. caprea x S. viminalis, and S. cinerea x S. viminalis). As some of the hybrids are of unknown origin (grown in experimental gardens, some material no longer available for morphological study!), samples with an unclear pattern are considered as unidentified taxa. Reviewer refers to "what about the other 41", but it is unclear what he is referring to. However, further explanatory details were added to Legend of Figure 1.

3) There really was a mistake in the total number of samples (comment on line 395). Thank you for noticing such an important issue. The Supplementary Table S1 was correct, but we forgot to mention additional samples from other project (has-myr) in the Methods section. This has now been corrected.